# Nordic Vets against AMR—An Initiative to Share and Promote Good Practices in the Nordic–Baltic Region

**DOI:** 10.3390/antibiotics11081050

**Published:** 2022-08-03

**Authors:** Susanna Sternberg-Lewerin, Sofia Boqvist, Simen Foyn Nørstebø, Thomas Grönthal, Annamari Heikinheimo, Venla Johansson, Viivi Heljanko, Paula Kurittu, Nils Fall, Ulf Magnusson, Ane Mohn Bjelland, Henning Sørum, Yngvild Wasteson

**Affiliations:** 1Department of Biomedical Sciences & Veterinary Public Health, Swedish University of Agricultural Sciences, Box 7036, 75007 Uppsala, Sweden; sofia.boqvist@slu.se; 2Department of Paraclinical Sciences, Faculty of Veterinary Medicine, Norwegian University of Life Sciences, Box 5003, 1432 Ås, Norway; simen.foyn.norstebo@nmbu.no (S.F.N.); ane.mohn.bjelland@nmbu.no (A.M.B.); henning.sorum@nmbu.no (H.S.); yngvild.wasteson@nmbu.no (Y.W.); 3Animal Health Diagnostics Unit, Finnish Food Authority, Mustialankatu 3, 00790 Helsinki, Finland; thomas.gronthal@ruokavirasto.fi; 4Department of Food Hygiene and Environmental Health, University of Helsinki, PL 66, 00014 Helsinki, Finland or annamari.heikinheimo@ruokavirasto.fi (A.H.); venla.johansson@helsinki.fi (V.J.); viivi.heljanko@helsinki.fi (V.H.); paula.kurittu@helsinki.fi (P.K.); 5Microbiology Unit, Finnish Food Authority, Mustialankatu 3, 00790 Helsinki, Finland; 6Department of Clinical Sciences, Swedish University of Agricultural Sciences, Box 7054, 75007 Uppsala, Sweden; nils.fall@slu.se (N.F.); ulf.magnusson@slu.se (U.M.)

**Keywords:** antimicrobial resistance, veterinary, education

## Abstract

In the Nordic countries, antimicrobial use in animals and the prevalence of antimicrobial resistance are among the lowest in Europe. The network “Nordic Vets Against AMR” organized a meeting in 2021, with key actors including representatives from universities, veterinary authorities and veterinary organizations in Finland, Norway and Sweden. This paper reflects the most important discussions on education, research, policy and future perspectives, including the experiences of these countries. It concludes that Nordic veterinarians are well placed to lead the way in the fight against antimicrobial resistance and that the sharing of experiences can support colleagues in other countries. Veterinary education must go hand in hand with research activities and continuously updated guidelines and legislation. There is also a need for postgraduate training on antimicrobial resistance and prudent antimicrobial use. The veterinary profession must, by any means necessary, protect the efficiency of antimicrobials for the sake of animal health, animal welfare and productivity, as well as public health. While restrictive use of antimicrobials is crucial, the ability of veterinarians to use this medical tool is also important for the sake of animal welfare and global food security.

## 1. Introduction

Antimicrobial use (AMU), here referring to the use of antibacterial drugs in animals, in the Nordic countries and the prevalence of antimicrobial resistance (AMR) are among the lowest in Europe [1,2]. This is the result of long-standing strategic work that is part of the Nordic tradition of prudent AMU. The issue of AMR is of vital importance to the veterinary profession, not only because of its public health responsibilities but also because it is vital for animal health.

The network “Nordic Vets Against AMR” was initiated by veterinary faculties in Finland, Norway and Sweden, based on funding by NKJ (the Nordic Joint Committee for Agricultural and Food Research), with the aim of allowing veterinarians in the region to share knowledge and experience of how AMR challenges can be met, and providing common ground for joint initiatives. To launch the network, a digital meeting was held in late 2021, attracting around 120 participants from the Nordic–Baltic countries (Finland, Norway, Sweden, Denmark, Iceland, Estonia, Latvia and Lithuania). The participants represented a number of key actors, including relevant veterinary authorities, national veterinary associations and veterinary faculties, and included veterinarians with different levels and fields of professional experience. Before the meeting, information about existing legislation, voluntary guidelines and policies and the veterinary curriculum in the Nordic–Baltic countries was collected. This paper reflects the most important meeting discussions related to education, research, policy and future perspectives, including some experiences in Finland, Norway and Sweden.

## 2. Topics

### 2.1. Education

Improving awareness and understanding of AMR through effective communication, education and training is part of the WHO Global Action Plan on Antimicrobial Resistance [3]. In the European Union, veterinary education is regulated by EU Directive 2013/55/EU. The quality and standard of veterinary medical establishments and their teaching are examined and accredited by the European Association of Establishments for Veterinary Education (EAEVE)/European System of Evaluation of Veterinary Training (ESEVT) at regular intervals.

According to a presentation by Carmen Espinosa-Gongora [4], veterinary students in Nordic countries have a high level of knowledge of AMR compared to other European countries. A striking difference has been demonstrated in the level of knowledge of the respective national guidelines and overall guidelines on prudent AMU. Whereas 70% of the European veterinary students were not aware of any AMU guidelines in animals, in the Nordic countries participating in the study (Denmark, Finland, Norway and Sweden), the percentage of students aware of the guidelines was 100%. Furthermore, in European countries, it seems that the level of knowledge of the guidelines is associated with the prevalence of AMR. This finding prompted the question of whether every European country has national guidelines and if these are taught in the veterinary curricula. The network suggested that the EAEVE accreditation should require the teaching of any national guidelines in the veterinary curricula.

In addition, the network wanted to promote lifelong learning and a changed mindset among veterinarians on the topics of AMR and AMU. This could be achieved by sharing experiences and strengths between countries in, e.g., webinars and courses, perhaps in collaboration between veterinary faculties and veterinary associations in different countries. Inter-Nordic topics could cover, for example, One Health and global AMR perspectives, success stories of prudent AMU and infection prevention. Furthermore, veterinary education in Nordic countries could involve skill building in inter-sectoral working, as tackling AMR requires close cooperation among professionals from various fields [5]. In addition to natural sciences, veterinary students could benefit from being introduced to the social sciences to improve their work in AMR prevention (e.g., behavioral sciences).

### 2.2. Research

To tackle some of the more complex issues related to AMR and AMU, a multidisciplinary, as well as a multisectorial (i.e., One Health) approach is needed. In addition, due to already existing structures and barriers between researchers and veterinary practitioners, there are challenges that need to be addressed.

To address research issues related to both AMU and AMR, a wide range of expertise is often needed—for example, researchers from life sciences and social sciences, clinicians, policymakers, advisory organizations and experts in framing research messages. To make such multidisciplinary collaborations successful, all parties must recognize their respective roles and competencies and acknowledge the time it takes to make multisectoral collaboration successful. The experiences that were shared demonstrated the need to recognize the different traditions and career-building steps within different research disciplines for multidisciplinary research to work well, for example, concerning research methodology and how research findings are published and disseminated.

More research is needed within several fields connected to AMR and AMU—for example, studies providing evidence for more precise and accurate individual antimicrobial treatments of different infectious diseases, infection epidemiology and studies on the contribution of different animal sources to AMR diffusion in the community.

Data availability and data sharing between different sectors within One Health are needed to effectively address complex research questions, for example, between agencies, institutes and universities. However, the data must not only be available; they also need to be of good quality and harmonized to allow their full use. There might also be issues with confidentiality and data security, but these must be resolved due to the detrimental long-term consequences of AMR. There are some open data sources, such as nucleic sequence databases and overall animal population data. However, detailed data on animal production systems, health and treatment records are only available based on specific agreements, if at all.

Research funding policies vary between different countries and consist of a mixture of calls where researchers can apply for any kind of research and calls addressing specific research themes. Researchers might perceive a lack of transparency when specific research themes are developed, and it might be that such themes are more politically driven than driven by research needs. This is of particular importance when it comes to research within AMR and AMU, since threats related to those areas are not so evident from a short-term perspective and might not cause alarming headlines, compared to topics such as multicontinental outbreaks of HPAI (avian influenza virus) and the COVID-19 (SARS-CoV-2) pandemic. Research on AMR and AMU needs to be addressed with a long-term perspective and a focus on proactive One Health aspects, which is not always taken into account in the current fragmented landscape of funding.

Another important issue is how stakeholders at local, regional, national and international levels can access science-based knowledge to address issues related to AMR and AMU in advisory services, policies and legislation. This is a matter of funding not only research, but also providing resources for the translation of results into science-based policies.

### 2.3. Legislative Framework

The Nordic countries are pioneers in regulation and policymaking on AMR and AMU. All three founding countries of the Nordic Vets Against AMR network (Finland, Norway and Sweden) have national strategies to combat AMR [6,7,8], as well as voluntary guidelines [9,10,11] for antimicrobial treatments of animals (when to treat, recommended substance, dosage, administration route and treatment time) and a national surveillance program for AMR and AMU [12,13,14]. These surveillance programs are conducted in collaborations between veterinary and public health authorities. This continuous surveillance enables veterinary officials and, by extension, policymakers to make informed and data-based decisions to drive AMR policy. The detailed description of the systematic data collection can also make it useful for research. In addition, all three countries classify at least some types of multiresistant bacteria as notifiable and have legislation banning the use of certain antimicrobials in animals, such as fourth-generation cephalosporins and carbapenems, and restricting the use of others, such as third-generation cephalosporins and quinolones.

Key points portraying the AMU and the legislative mentality in the Nordic countries were noted in a recent paper on the Nordic approach to bovine udder health [15]. A common theme was that antimicrobials can only be obtained by veterinary prescription. Thus, the legislation puts the onus on veterinarians to decide whether antimicrobials are needed or not. The use of antimicrobial substances that are critically important for human medicine is discouraged, and voluntary guidelines support veterinarians in the use (or non-use) of antimicrobials for the most common infections. These guidelines recommend the primary use of non-antimicrobial treatments. If antimicrobials are deemed necessary, narrow-spectrum substances should be the first choice, whenever possible, and the use of broad-spectrum medication should usually be reserved for life-threatening illnesses, guided by susceptibility testing. Veterinarians are also held accountable by veterinary regulatory agencies for their prescribing of antimicrobials, and they are not allowed to make a profit from the sales of pharmaceuticals. Instead, income is generated from providing other animal health services, such as disease prevention.

Estonia and Lithuania have also devised national strategies for combatting AMR [16,17]. Both include specific actions for key actors to impact the AMU and AMR. These actions include, among other things, educating both professionals and the public, the establishment of task groups and specific targets for reduced AMU and AMR.

As in all EU member states, the use of antimicrobials as growth promoters has been banned in the network’s founding countries. The EU-wide ban came after an initiative from Sweden, where antimicrobials as feed additives were banned in 1986. In addition, the routine use of antimicrobials for prophylaxis was prohibited in the core countries, even before the new EU Veterinary Medicines regulation (EU 2019/6) was developed. The legislation and voluntary guidelines in the founding countries combine to make susceptibility testing before antimicrobial treatment and the use of narrow-spectrum antimicrobials, whenever possible, part of the veterinary mentality. This has been reflected in a study that was presented by Nancy De Briyne from the European Veterinary Federation (FVE) [18]. It appears that the legislative framework supports Nordic veterinarians in their AMU and is not perceived as too restrictive.

## 3. Discussion

Like all microbes, resistant bacteria—commensal or pathogenic—do not respect national borders. Thus, even if the current situation in the Nordic countries is favorable, a global perspective must be applied to the AMR issue, in animals and in humans. Hence, in parallel with the continuous efforts to reduce AMU in animals, Nordic veterinarians should share the success factors that have made it possible to rear healthy and productive animals under good animal welfare conditions with low use of antimicrobials [19,20,21]. In a global context, the conditions for managing a prudent and medically rational use of antimicrobials are mostly very different from those in the Nordic countries. In low- and middle-income countries (LMICs), access to qualified veterinary services may be inadequate, veterinary pharmaceuticals are readily available without prescription and consumer awareness is often low. Even if there is legislation on the prescription of pharmaceuticals and bans on using antimicrobial feed additives, resources to enforce these may be lacking [22]. In these countries, the Nordic experiences of collaborative work on good animal husbandry, efficient biosecurity and adequate vaccination programs to keep animals healthy, and thereby reduce the need for antimicrobials, could be shared with colleagues and animal owners.

In an EU context, where legislation related to antimicrobials is already in place, the Nordic vets could promote a veterinary service more oriented towards preventive animal health, thereby reducing the need for AMU. This may require improved veterinary skills in biosecurity and vaccinology and other disease-preventive measures, in veterinary curricula or in continuing education. Additionally, EU legislation preventing veterinarians from earning revenue linked to the sale of antimicrobials should be introduced. Globally, a requirement for access to antimicrobial drugs based on veterinary prescription only should be strived for. Nordic vets participate in the COST Action CA18217 European Network for Optimization of Veterinary Antimicrobial Treatment (ENOVAT), which focuses on guidelines for treatment and diagnostic procedures. In this context, the European Veterinary Committee on Antimicrobial Susceptibility Testing (VetCAST) will also contribute to global standards for antimicrobial susceptibility testing of bacterial pathogens of animal origin.

The Nordic countries have had surveillance programs for AMR and AMU within the human, animal and food sectors for more than two decades. These data have been utilized for policy and research purposes but, as the Nordic countries are also unique in their extensive databases for monitoring animal health, there is a potential to combine and analyze these data. In addition, new data collection opportunities provided by automatic livestock management systems and new analytical tools within artificial intelligence offer new prospects to understand the dynamics of AMR transmission from farm to fork.

As AMR is driven by AMU, an obvious strategy would be to explore novel ways of reducing the occurrence of bacterial infections. The development of vaccines against bacterial pathogens is advancing, but there is a gap in the functional effectivity of some of these. In addition, research on how to optimize animal husbandry and breeding for healthy animals is needed. This includes aspects of the normal microbiota and immune system of the individual animals, as well as factors such as reducing social and other stress factors, optimizing feed contents and feeding frequency and targeting the use of disinfection routines to key points in the production operation to maintain protective microbiomes.

In the Nordic countries, the national veterinary and public health authorities collaborate to complement the WHO Global Action Plan on AMR and the EU One Health Action Plan against AMR. The need for future environmental surveillance is currently being discussed among these partners. Although the environment is the most dynamic and complex sector of the three One Health pillars, the environment has, so far, often been the neglected part of AMR initiatives. From the AMR perspective, the environment is an indirect transmission route for resistant bacteria and resistance genes. More emphasis is now put on understanding the relative roles of the human, animal and environmental components in the development, transmission and persistence of AMR. There is increasing activity within the Nordic countries in terms of developing and discussing methods for environmental surveillance, particularly the use of wastewater as a reflection of the local human population microbiome, AMR determinants and drivers. A deeper understanding of the environmental impact of different AMUs (substance, dosage, route of administration, treatment period, etc.) is needed to manage AMR in a true One Health manner.

The data presented in the meeting indicate that veterinary students in the Nordic countries have a high level of knowledge of AMR compared to other European countries. This does not mean that the Nordic veterinary faculties can afford complacency. On the contrary, they should keep improving the relevant components of their curricula and sharing their experiences. In addition, continuous professional development activities should be conducted; collaboration between Nordic veterinary faculties may be a cost-efficient way of ensuring access to such training.

Competence is a concept that integrates knowledge, skills and attitudes. Overall basic veterinary competence is currently laid down in different pieces of EU legislation, of which the Directive 2005/36/EC (amended by Directive 2013/55/EU) is crucial. Based on this legislation, the EAEVE has “translated” the requirements to the more specific list of Day One Competences; the minimum standard required for veterinary teaching and the starting point for the veterinary profession and the Nordic veterinary faculties should promote more emphasis on competences within AMR and AMU for all European veterinary students.

The veterinary profession must protect the efficiency of antimicrobials to ensure animal health, welfare and productivity, as well as public health. While restrictive AMU is crucial, the ability of veterinarians to use this medical tool is also important, for the sake of animal welfare and global food security.

## 4. Conclusions

Nordic veterinarians are well placed to lead the way in the fight against AMR; sharing their knowledge and experience can support colleagues in other countries.

Veterinary education must go hand in hand with continuous research activities and continuously updated guidelines and legislation. The EAEVE accreditation should require the teaching of any national guidelines in the veterinary curricula.

There is also a need for postgraduate training on AMR and prudent AMU.

## Data Availability

Not applicable.

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
