# Peer review of "Nordic Vets against AMR—An Initiative to Share and Promote Good Practices in the Nordic–Baltic Region"

_antibiotics, 2022, doi:10.3390/antibiotics11081050_

Round 1

Reviewer 1 Report

The manuscript intitled "Nordic Vets Against AMR – an initiative to share and promote good practices in the Nordic-Baltic region" although very well writeen andpresented, and the authors showed a great improvement of the manuscript since the last review do not bring any thing new to the community. In my opinion is not suitable yo be publish in this journal.  Does not bring anything new to the scientific knowlege and is not suitable to the scopus of this journal. 

Author Response

As the reviewer's only concern is that the manuscript does not bring any new scientific knowledge, we are not able to provide any changes to meet this demand. We agree that there might not be new scientific knowledge in the paper but we would argue that we compile evidence of successful approaches to antibiotic stewardship on national level that is of use for the journal's readership, and as the manuscript is submitted as Perspective, not an original scientific article, it does fit the scope of the journal. However, this is of course up to the editor to decide.

Reviewer 2 Report

Grammar and syntax can still be improved. 

Author Response

We have read through the manuscript and checked with US English Grammar and spelling and made some minor edits to improve the language.

Reviewer 3 Report

This is a well written manuscript. However, there is hardly a mention of challenges faced by the baltic countries in terms of AMU and AMR in animals and how the experience in the Nordic countries could specifically contribute to address. I am also intrigued by the fact that all authors are from Nordic countries and NONE is from the baltic region.

The manuscript seem to use Antimicrobials and antibacterial agents rather interchangeably, creating confusion to the readers

Author Response

As described in the text, the authors are the original founders of the network. Veterinarians from the Baltic countries, and Denmark, were invited to join the meeting and the network and we will continue to work on this collaboration. We have included the information provided by the Baltic participants, but a broad and in-depth knowledge of the situation and activities were only available from the founding countries (Sweden, Finland and Norway).

We have revised the entire text as regards the use of antimicrobials and antibacterial agents, thank you for this comment.

Reviewer 4 Report

This article expresses the opinion of a group of opinion leaders  from northern European countries where the use of antimicrobials (AMD) is effectively better controlled than elsewhere. I have very little to say about the opinion of a group which does not hide a reasonable self-satisfaction, but one can be surprised at the following points (i) there is no discussion of the need to develop new so-called eco-friendly AMDs in the section on environmental impact. It is correct  to say that the environment is a real hub in the exchange of genes of resistance between human and  veterinary  medicine and also the different ecosystems including plants and vegetables. In this respect,  all antibiotics have not the same impact on the intestinal animal microbiome and on their persistency in the environment can be very different. it is true that talking about the marketing of a new AMD in veterinary medicine is a real regulatory taboo, while spreading new generics that take up completely obsolete dosages with the aim of lowering prices does not seem to be a regulatory difficulty (ii) the issue of validation and harmonization at EU level of Antimicrobial Susceptibility Testing (AST) is not mentioned even though this is one of the weak points of the prudent use of AMD. This is  because prescribers are dubious about their clinical relevance due to the absence of any field validation of these tests (see de Bryne's article). it is true that the microbiologists who manage the laboratories carrying out these AST hardly insist on this point . As such, the VetCAST / ENOVAT project would have deserved to be mentioned insofar as it aims to fill this gap.  (iii) the problem of dosages which should be revised upwards for many AMDs is not mentioned and regulatory organizations likely   prefer to let prescribers hypocritically take on this responsibility, especially for tetracyclines in food producing animals and amoxicillin in dogs. (v) finally, reading this document and looking at the CVs of the authors, it seemed to me to reflect mainly the opinion of microbiologists and epidemiologists. I think that a real progress would also be to ask opinion of  Industry  that  develop (or are no longer motivated to develop) new innovative antibiotics for veterinary medicine with the goal to mitigate the different one health issues and   sociologists, economists, pharmacologists and field practitioners to better understand some issues rightly  raised in this document .

Author Response

We appreciate the valid points made by the reviewer and have tried to address them as best we can.

We refrained from discussing the issue of new drugs because this did not come up in the meeting and the trend in current EU policy discussions lean strongly towards increasing the restrictions on veterinary use of antimicrobial drugs, making it unlikely that any new antimicrobials would ever be allowed for use in animals. As for the generics, if misuse can be addressed this will also address the issue of overuse of cheap generics. However, as the environmental issue appears not to have been sufficiently emphasised in the manuscript, we have added a sentence on lines 246-248 to reflect this.

As for the De Briyne paper, we discovered a mistake in the reference list which has now been corrected. The reasons for not using AST (given as free text responses in that study) included not only doubts about validity but “sampling difficulties, the urgency of the situation, concerns regarding the clinical relevance of in vitro tests and owners being unwilling to pay for such testing” and we did not delve into any of these reasons. Nevertheless, we appreciate the point about VetCAST and ENOVAT and have added a reference to them on lines 213-218.

Treatment guidelines generally include advice on substance, dosage, route of administration and treatment period. We have clarified this on lines 140-141. Most of the existing guidelines that we describe in our countries include science-based advice on dosages and hence this is not left to individual prescribers.

As for the background of the authors being mainly academics, the network includes many other competencies and the need for multidisciplinary research efforts is highlighted in paragraph 2.2. Research. We are hesitant to add any suggestions about how to involve the pharmaceutical industry as the focus of our paper is on veterinarians and how we can share and promote good veterinary practices to tackle AMR.

This manuscript is a resubmission of an earlier submission. The following is a list of the peer review reports and author responses from that submission.

Round 1

Reviewer 1 Report

This manuscript is a short review or meeting summary on the current practice of Nordic veterinarians and related professionals to address and prevent/ameliorate antimicrobial resistance by the prudent use of antimicrobials or primarily antibiotics. The manuscript provided some useful or interesting information, but is often very repetitive/redundant, unnecessarily verbose, often colloquial, and non-specific. Furthermore, it often states the obvious and  is not well-written with various syntactical errors and poor grammar that could have been corrected using the spelling/grammar tool on Microsoft Word.

The review seems to concentrates only on livestock or farm animals, without reference to small/companion animals. poultry or zoo/exotic animals.

Only a few examples of the poor grammar, etc. follow:

Line 76: One Health not One health;

Lines 91-92: Researchers often have the most updated knowledge of AMR mechanisms and drivers (use causes)  thereof, however, the, although clinicians are the ones that are closest to AMU in the primary users of antimicrobials due to their contact with animal owners/farmers.

Lines 94-98: The authors often have a sentence and then add a comma followed by "for example", which is incorrect.

To address research issues related to both AMU and AMR, (add comma here) a wide range of expertise is often needed,  for example including researchers from life sciences and social sciences, clinicians, policymakers, advisory organizations and experts in framing research messages communication. To make such multi-disciplinary collaborations successful, all parties must recognize their respective roles and competences expertise and also acknowledge the time it takes to make multi-sectorial collaboration successful.

Lines 99-102: The following sentence is very poorly written and can be condensed as follows:

Experiences shared Shared experiences underscored the need to recognize the different traditions and career-building steps within different research different progressive acquisition of professional skills of various disciplines that eventually result in the success of multi-disciplinary research. multi-disciplinary research to work well, for example concerning research methodology and 101 how research findings are published and disseminated.

Lines 103-107: Research is lacking within several fields connected to AMR and AMU, for example including studies providing evidence for investigating more precise and accurate individual antimicrobial treatments of different infectious diseases, use of advanced molecular techniques for studies on dissemination of emerging AMR determinants and infection epidemiology of infection, and studies on the contribution of different animal sources to AMR diffusion in the community.

Lines 108-113: Data availability and data sharing are needed necessary to effectively address complex research questions, for example between among agencies, institutes and universities. However, the data must not only be available, it also needs to be of good quality and harmonized (poor choice of words; use a different better word) to allow full use of it for efficient utilization of the data. There might also be issues with confidentiality and data security, but these should be regarded as technical issues possible to solve due to the enormous long- 112 term consequences of AMR on animal and public health.  that should be easily resolved.

Please provide a brief discussion of potential privacy issues. Specifically, what are the potential issues and why might they occur?

Lines 126-132: Research funding policies vary between different among countries and might consist of a mixture between calls where researchers can apply for any kind of research and calls addressing specific research themes.  announcements for specific and/or non-specific research topics.

Lines 130-136: This is of particular importance when it comes to research within AMR and AMU, (add comma) since threats related to those areas are not so evident in the short-term perspective and might not cause alarming headlines, compared with for example multi-continental outbreaks of HPAI (highly pathogenic avian influenza virus) and the current Covid-19 (SARS-Cov-2) pandemic.

Lines 134-136: This is one example of how the authors not only state the obvious.

Research on AMR and AMU needs to be addressed in a long-term perspective as it silently but severely affects the health of animals and humans.

Lines 143-145: This continuous surveillance enables veterinary officials and, by extent, policymakers to make informed and data-based decisions to drive AMR policy.

Lines 151-153: The AMR situation of the Nordic countries with regard to bovine mastitis has recently been explored [15]. In this report some key points which can be generalized to cover the AMU in the Nordic countries and portray the legislative mentality (this is a poor word; choose another word) were noted.

Lines 169-174: An example of poor sentence structure

In addition, the routine use of antibiotics for prophylaxis was prohibited in the core countries even before the new EU Veterinary Medicines regulation (EU 2019/6) was developed. The legislation and voluntary guidelines in the core countries combine to make susceptibility testing before antibiotic treatment and the use of narrow-spectrum antibiotics whenever possible part of the veterinary mentality.

Line 178: This sentence not only states the obvious, but is too colloquial. Re-write or delete.

Like all microbes, resistant bacteria, commensal or pathogenic, don’t respect borders.

Lines185-187: In low- and medium-income countries, where access to qualified veterinary service may be poor limited, antibiotics are readily available without prescription and consumers consumer awareness is usually low. Even if there is legislation on prescription of pharmaceuticals and bans of using antibiotics as a feed additive, resources to enforce these regulations may be lacking.

Lines 195-198: This may require improved veterinary skills in knowledge of biosecurity and vaccinology and, but but also additional methods of disease prevention measures, in included in veterinary curricula or offered as continuing education. Also, legislation preventing veterinarians from earning a revenue linked to the sale of antibiotics should be introduced.

Line 205: new prospects to understand the dynamics of AMR transmission from farm to fork at any point within the production system.

Line 214: routines to at key points in the production operations.

Lines 226-227:  ... environmental surveillance, particularly the use of wastewater as a reflection of the society.

Wastewater as a reflection of society!? Very poorly written, please re-write!

Example: Analysis of wastewater to determine the concentration of antibiotics and/or the presence of resistant bacterial organisms.

Line 232:  ... faculties can sit back and relax  become complacent. (Be more professional in writing.)

Lines 237-239: Competence is a concept that integrates knowledge, skills, and attitudes. Overall basic veterinary competence is currently laid down included in different pieces of the EU legislation, of which the Directive 2005/36/EC amended by Directive 2013/55/EU is crucial.

Line 251: Nordic veterinarians are well placed to lead the way in the fight (This is not a fight; again, too colloquial. Use a better and more professional word) against AMR and ...

Reviewer 2 Report

The manuscript Nordic "Vets Against AMR – an initiative to share and promote good practices in the Nordic-Baltic region", is an interesting perspective that gives a view of the knowledge of AMR in north Europe. It is based on a meeting with a relatively low number of people to extrapolate in general. Although interesting does not bring anything new to the field.

It is very well written but is quite extensive for a perspective paper. 

Reviewer 3 Report

The manuscript "Nordic Vets Against AMR – an initiative to share and promote good practices in the Nordic-Baltic region" focus on a very important topic and provides ideas and examples to be used and applied in other countries.

The structure of the manuscript is adequate, is easy to read and understand. All sections are relevant and the content can be very useful to universities and other veterinary organizations in the European Union.